# Analysis and Comparison of Role-Based Interorganizational Workflows for a Construction Project

**Jianlin Jiang [1,2,\*], Jianguo Chen [1], Rongyue Zheng [2] and Yan Zhou [2]**

[1] School of Economics and Management, Tongji University, Shanghai 200092, China
[2] Faculty of Architecture, Civil Engineering and Environment, Ningbo University, Ningbo 315211, China
\* Correspondence: jiangjianlin@nbu.edu.cn; Tel.: +86-139-5820-7405

**Abstract:** The implementation process of construction projects is an iterative process of continuous modification and improvement among participant organizations. Traditional workflow analysis methods for a single organization are not suitable for the analysis of such implementation processes. Therefore, an interorganizational workflow analysis method based on organizational roles and associated with their collaborative relationships is required. In this study, a role-based interorganizational workflow model for participant organizations is developed, in which it is assumed that interoperability has a loosely coupled form for temporary multi-organizations. The Fuzzy Analytic Hierarchy Process (FAHP) is applied to determine the parameters of the correlation between interorganizational workflows, which includes downstream sensitivity and the probability of change. Furthermore, according to workflow interactions between organizations, an analysis model of interorganizational workflow is developed by using the Design Iteration Model for reference to analyze the time performance of participant organizations. Additionally, two forms of interorganizational workflow are compared and analyzed. Some suggestions are put forward to improve interorganizational workflow management and reduce the total time taken to complete the workflow processing of each organization (T) and the total time spent on the interorganizational workflow process (effort, E). This research may help strengthen interorganizational workflow management and enrich the workflow modeling theory.

**Keywords:** construction project; temporary multi-organizations; interorganizational workflow; loosely coupled

---

## 1. Introduction

Construction projects are carried out by temporary teams of heterogeneous organizations called temporary multi-organizations (TMOs) [1–4]. During construction projects, many participants from different organizations should cooperate temporarily [5]. For example, such projects involve many participating organizations, such as the owner, designer, constructor, supplier, general contractor, subcontractor, operator, and more. Organizational complexity in projects is the degree of operational interdependency and interaction between the project organizational elements [6,7]. Reciprocal dependencies represent the highest level of complexity and dominate the construction process [8]. The works of different participant organizations in a construction project are constrained and conditioned by each other. With mutual restriction and the interaction between many factors, complex workflow relationships are formed among organizations.

Implementation processes for construction projects, such as the planning and design phase, do not follow a simple linear progression between organizations, but rather undergo an iterative

process of continuous modification and improvement among TMOs. Repeated iteration is an important feature of TMO integration. For example, the detailed design process for prefabricated buildings involves iterations of successive revision, similar to agile development [9]. Planning and design often fail, and repeated multi-organizational design is a way to find effective and satisfactory solutions. The integration process of TMOs also involves repeated interorganizational redesign. When interorganizational conflict has not been eliminated, conflict resolution should be carried out. TMO integration is no longer a simple and repetitive process of design modification but is a process of redesign. A complex feedback loop is formed among the workflow subsystems of TMOs. Dependence, coupling, and interactive workflows among organizations would lead to a large number of interorganizational reworkings and work iterations.

Workflow systems have significant temporal aspects. Activity sequencing, deadlines, routing conditions, and complex scheduling constraints all involve the element of time [10]. There are many constraints among participant organizations and overlap and conflict in the interorganizational workflow are very frequent. Due to the interorganizational interactive workflow and loosely coupled interdependence among TMOs, the workflow processing time of each organization is difficult to analyze. For each participant organization of a TMO, it is important to study how to effectively analyze performance information, such as the time or workload of the interorganizational workflow.

Interorganizational workflows can be analyzed from three perspectives: activity-based, information processing-based, and role-based. In role-based interorganizational workflows, interorganizational relations are regarded as organic social systems, which are composed of relationships among member organizations. "Role" is an abstraction of a project participant with a set of skills, and reflects the responsibilities of individual organizations. However, the implementation process of an interorganizational workflow reflects interdependence and execution ordering between activities. It essentially reflects interdependence between the organization of activities. The source of this dependence is that one-member organization controls the results of the activities of another member organization or exchange resources [11].

Current research on interorganizational workflows is mainly conducted in computer-based fields such as e-commerce. Such research is mainly technical, task-oriented micro-analysis [12–15]. Research on interorganizational workflows has the following deficiencies [16].

(1)  Most workflow models focus on the process perspective and neglect the organizational perspective [17,18].
(2)  Workflow analysis is aimed at the verification of reliability and validity [19–21], while research on quantitative performance analysis is less likely to be aimed at such a verification [22,23].
(3)  Such research ignores collaborative relationships between participants.
(4)  The factors of reworking and iteration caused by reciprocal workflow or interdependence and coupling among organizations are not involved in such research.

Research on the scheduling for construction projects and new product development (NPD) usually focus on a single organization and a multidisciplinary environment, while the background of TMO ecology is ignored [24,25]. Traditional workflow analysis methods, which are suitable for single organizations, are difficult to adapt to TMO ecology. Such adaptation requires an interorganizational workflow analysis method, which is based on organizational roles and associated with the collaborative relationships between organizations.

This paper studies the interorganizational workflow of TMOs from the perspective of organizational roles and their interaction. The research aims include the following:

(1)  the development of an analysis model of role-based interorganizational workflow,
(2)  the calculation of time or workload performance indicators via modeling considering interorganizational reworking and iteration phenomena,
(3)  the analysis and contrast of interorganizational workflows with different forms of interoperability, so as to determine an appropriate form of interoperability for interorganizational workflows,

(4)    and the discussion of measures to improve interorganizational workflow management.

This study conducts a modeling-based analysis of interorganizational workflows of TMOs in construction projects from a role-based perspective and calculates performance indicators such as time or workload by modeling. Interorganizational workflows in a loosely coupled form are compared with interorganizational workflows in a case transfer form. Moreover, some suggestions are put forward to improve interorganizational workflow management.

## 2. Modeling of Role-Based Interorganizational Workflows for Construction Projects

Temporary multi-organizations of construction projects operate in a dynamic, distributed environment. Workflows of all participating organizations are integrated into multi-organizational workflows in the project ecosystem by a cooperative mechanism. Each participant organization interoperates in the interorganizational workflow using a Building Information Modeling (BIM) collaborative work platform (an interorganizational information system) to integrate TMOs. Taking a prefabricated building project as an example, project planning and design needs to consider prefabrication, transportation, assembly, and other requirements as a whole, and also needs to consider all kinds of pre-embedded and reserved designs for prefabricated components so as to eliminate problems that occur in the manufacturing and installation process in advance. Therefore, it is necessary for all participant organizations to cooperate closely with each other. Multiple participant organizations coordinate with the interorganizational workflow based on the Building Information Modeling collaborative work platform. Through multi-organizational cooperation and interorganizational workflow interoperation, component problems can be modified iteratively before manufacture or installation using a BIM integrated model so as to achieve efficient coordination among component design, factory manufacturing, and site installation. Additionally, negotiation and coordination with cooperative work can reduce the time and effort involved in communication among participating organizations.

### 2.1. Loosely Coupled Interoperability

In a construction project ecosystem, multi-organizational collaboration adopts a point-to-point collaborative structure of a loosely coupled workflow based on the parallel and interactive operation of workflow for each organization. The overall tasks of the construction project are executed concurrently by multiple participating organizations. Due to the various workflow dependencies among organizations, each organization often does not understand the impact of its own decisions on other organizations. Additionally, each organization needs feedback from other organizations to adjust its own decisions so that the construction design can continue to progress toward success.

Five forms of interoperability between workflows are summarized, which include capacity sharing, chained execution, subcontracting, case transfer, and loosely coupled interoperability [26,27]. The forms of interoperability of interorganizational workflows of TMOs among construction projects are different than the forms of interoperability of interorganizational workflows in computer-based fields. In order to investigate the phenomena of an interorganizational rework and iteration caused by interdependence and coupling among organizations, this paper mainly focuses on two forms of interoperability, which include case transfer and loosely coupled interoperability.

Of these five forms, loosely coupled interoperability is the most complicated, has the strongest dynamic, and is the most appropriate for process implementation in interorganizational situations [16]. In loosely coupled interoperability, the interorganizational workflow can be decomposed into several parts that can be operated concurrently by different organizations. Each part is relatively independent and is managed by one organization. At the same time, as a part of project ecosystem workflows, each part collaborates with other parts to achieve some common goals of the project. The concurrency and dynamics of interorganizational workflows of TMOs in construction projects make workflow management more complex when compared with the workflow of a single organization.

BIM collaborative work platforms, such as the "BIM Collaborative Platform of Project Management" developed by Glodon Company, Ltd., China, can achieve the integration and cooperation of participating organizations including the owner, designer, fabricator, and erector [5]. Based on a BIM collaborative work platform, the interoperability of the interorganizational workflow of TMOs of a construction project should be loosely coupled, and each organization works in parallel. As shown in the flow chart of a loosely coupled interorganizational workflow of TMOs within a construction project (Figure 1), there is no sequential connection of several internal processes of participant organizations, but rather a loosely coupled interaction of multiple roles (organizations).

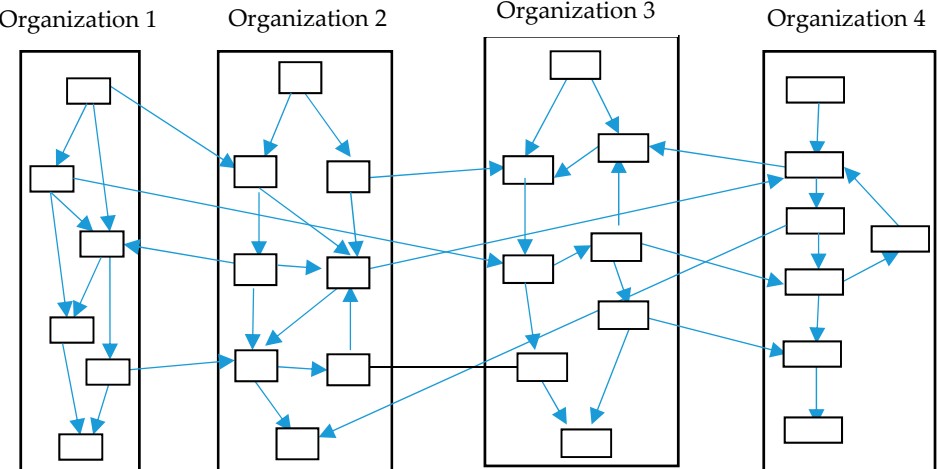

**Figure 1.** The loosely coupled form of workflow interoperability.

## 2.2. Model of Role-Based Interorganizational Workflow

The execution of an interorganizational workflow is divided into two parts: intra-execution, i.e., execution within the same organization, and inter-execution, i.e., interaction between organizations [28]. Similarly, the interorganizational workflow of TMOs in construction projects can be divided into two parts: the intraorganizational workflow and the interorganizational workflow. Both the intraorganizational and interorganizational workflows include pooled, sequential, reciprocal-compatible, and reciprocal-contentious workflows [29]. The modeling of intraorganizational workflows is relatively complete. However, more work remains to be done on the modeling of interorganizational workflows, and this will be the main direction of future research in this field. The modeling of interorganizational workflows emphasizes workflow links among organizations and the coordination of their interactions [16].

In this paper, interorganizational workflow is analyzed at the organizational level, taking the participating organization as the role. Given the enormous number of tasks performed by each organization and the workflow iteration among organizations, it is difficult to implement modeling and analysis for specific tasks and it is not easy or convenient to understand the level of participating organizations. Therefore, in the present study, an interorganizational workflow is analyzed by merging modeling from the role-based perspective of participating organizations.

The interorganizational workflow analysis, which is conducted in this paper, does not include intraorganizational workflows, which are independent of other organizations, such as pooled interorganizational workflows. Additionally, this analysis only considers workflows that are related among organizations, such as sequential interorganizational workflows, reciprocal-compatible interorganizational workflows, and reciprocal-contentious interorganizational workflows. Intraorganizational workflows that are independent of other organizations can be analyzed and calculated separately, since there is no association or iteration with workflows of other organizations. Intraorganizational workflows were not involved in the analysis and calculations performed in this paper.

In Reference [30], the concept of workflow merging was described and a systematic method was proposed to merge multiple simple workflows into a single complex workflow. In the present study, the part of the intraorganizational workflow for each organization, which is associated with other organizations, is considered to be a "work set" of the interorganizational workflow. Taking interorganizational workflow processing in the detailed design stage of a prefabricated building project as an example, participating organizations include the designer's prefabricated building design team (Organization 1), the fabricator's component design team (Organization 2), the transportation enterprise's transport planning team (Organization 3), and the construction contractor's construction planning team (Organization 4) (see Figure 2). In the analysis of the role-based interorganizational workflow, the "work set" of each organization that is associated with other organizations is regarded as a "task," and the relationships between the interorganizational workflows of different organizations simplify into the relationships between organizations. This simplification is expressed by $C_{j-i}$, i.e., there is a relationship between the workflows of Organization j and that of Organization i. The interorganizational workflow process is equivalent to parallel iteration among different "tasks," i.e., parallel iteration of "work sets" among organizations. The design results are optimized by repeated iterations.

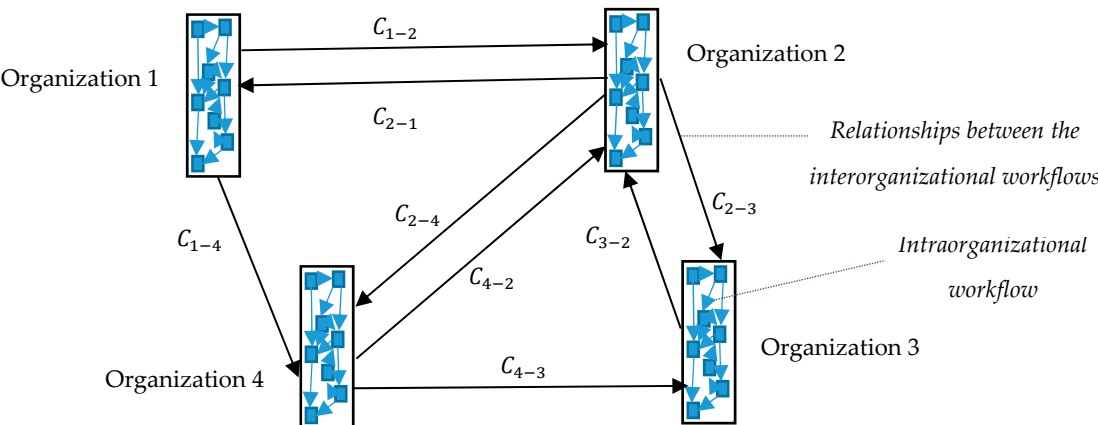

**Figure 2.** Role-based interorganizational workflow model.

## 3. Research Methodology

In this study, a loosely coupled interorganizational workflow iteration model was developed, which referred to the Design Iteration Model (an engineering design model). This was achieved by taking organization as a role and aiming at interorganizational rework and iteration phenomena caused by interactive workflows together with interdependence and coupling relationships among participating organizations. The analysis was based on each participating organization.

The Design Iteration Model was developed using the Design Structure Matrix (DSM) [31]. This model assumes that all coupled tasks are executed simultaneously in parallel and that the iteration workload is constant. Iteration-driven tasks in the design process are identified by establishing a Work Transfer Matrix (WTM) model. The Design Iteration Model was used for reference in the analysis of a role-based interorganizational workflow of a construction project.

*3.1. Correlation Parameters for Interorganizational Workflow*

3.1.1. Correlation Parameters

Taking the detailed design phase of a prefabricated building project as an example, the correlation between the interorganizational workflows of TMOs can be expressed by a Binary Design Structure Matrix (BDSM), as shown in Figure 3. A BDSM qualitatively reflects the existence of correlation between interorganizational workflows. However, it does not reveal differences in workflow correlation or the

strength of workflow correlation. In order to reveal the possibility and impact of correlations between interorganizational workflows, correlation can be described by a numerical Design Structure Matrix (NDSM). In this paper, the downstream sensitivity ($s_{ij}$) and probability of change ($c_{ij}$) are used to quantitatively express correlations between interorganizational workflows. The correlation $C_{j-i}$ is illustrated in Figure 4.

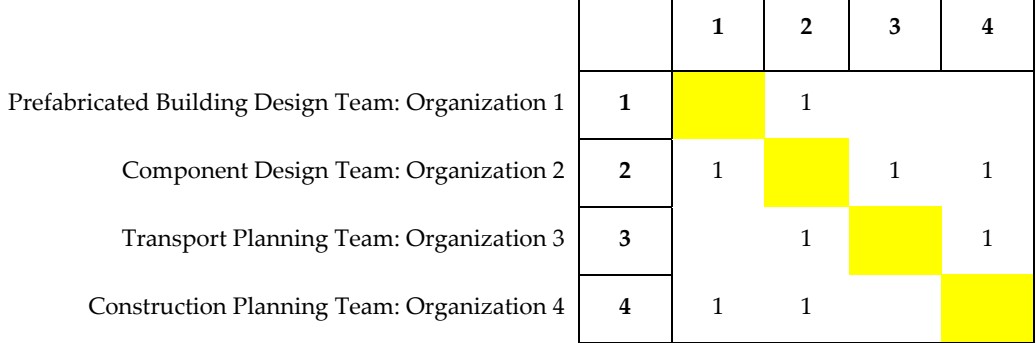

|  |  | **1** | **2** | **3** | **4** |
|---|---|---|---|---|---|
| Prefabricated Building Design Team: Organization 1 | **1** |  | 1 |  |  |
| Component Design Team: Organization 2 | **2** | 1 |  | 1 | 1 |
| Transport Planning Team: Organization 3 | **3** |  | 1 |  | 1 |
| Construction Planning Team: Organization 4 | **4** | 1 | 1 |  |  |

**Figure 3.** The Binary Design Structure Matrix used to determine the correlations between the workflows of different organizations.

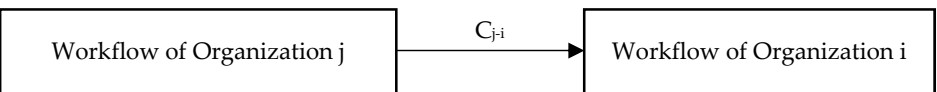

**Figure 4.** Correlation of interorganizational workflow.

(1) Downstream sensitivity

Downstream sensitivity ($s_{ij}$) refers to the degree of sensitivity (dependence) of the workflow of Organization *i* (downstream) to the processing workflow of Organization *j* (upstream). High sensitivity indicates that the processing workflow of Organization *i* is highly dependent on the workflow processing of Organization *j*, and that adjustment to the workflow of Organization *j*, therefore, leads to a high degree of adjustment to the workflow of Organization *i*. The opposite is true for low sensitivity.

(2) Probability of Change

The probability of change ($c_{ij}$) refers to the change in the range of the iterative workflow processing of Organization *i* (downstream) that is caused by a change in the range of the iterative workflow processing of Organization *j* (upstream). High changeability means that a change in the iterative workflow processing of Organization *j* would lead to a large range of iterative workflow processing for Organization *i*. The opposite is true for low changeability.

The correlation for the interorganizational workflow is comprehensively reflected by the two aspects of downstream sensitivity and probability of change. On this basis, these two aspects are transformed into a single comprehensive indicator, $r_{ij}$, to represent the correlation for the interorganizational workflow processing. In this paper, the product of downstream sensitivity and probability of change is used to represent $r_{ij}$. For an n-dimensional NDSM, R, we have the following.

$$R = r_{ij} = \begin{cases} s_{ij} \cdot c_{ij}, & (i \neq j) \\ 0, & (i = j) \end{cases} \tag{1}$$

3.1.2. Determination of The Correlation of Interorganizational Workflows

It is difficult to determine the two parameters $s_{ij}$ and $c_{ij}$. To achieve this task, the Fuzzy Analytic Hierarchy Process (FAHP) is used in this paper. The BDSM for the interorganizational workflow

correlation shown in Figure 3 is transformed into a numerical Design Structure Matrix (NDSM), as follows.

Step 1: A pairwise comparative Fuzzy Consistent Matrix of downstream sensitivity is constructed for the baseline workflow processing of the organization. The baseline workflow is the part of the intraorganizational workflow of each organization, which is associated with other organizations.

In this method, every element in the rows of the BDSM except the diagonal elements is used as a baseline workflow, and the other non-zero elements of the rows are pairwise compared to determine the importance of the elements to the baseline workflow. In Figure 5a, $X_r$ denotes the baseline element of the $r$th row ($\forall r = 1, 2, \cdots, n$). $\{X_i \cdots X_j\}$ are the non-zero elements of the $r$th row ($\forall i, j = 1, 2, \cdots, n$) and $x_{ij}$ is $X_i$ compared with $X_j$ considering the importance of $X_r$, where $x_{ji} = 1 - x_{ij}$. The principal eigenvectors of each Fuzzy Consistent Matrix, i.e., the n-dimensional weight vectors, are calculated. Then, the NDSM of the downstream sensitivity is constructed.

(a)                                                    (b)

**Figure 5.** Fuzzy consistent matrices of downstream sensitivity (**a**) and probability of change (**b**).

Step 2: The pairwise comparative Fuzzy Consistent Matrix of the probability of change is constructed for the baseline workflow processing of the organization. This method is similar to that in Step 1, except that column comparisons are performed instead of row comparisons, as shown in Figure 5b. Lastly, the NDSM of the probability of change is constructed.

Step 3: The final NDSM matrix, R, for the interorganizational workflow correlation is obtained by multiplying the elements corresponding to the NDSM of downstream sensitivity and the NDSM of the probability of change.

*3.2. Analysis of Role-Based Interorganizational Workflow*

3.2.1. Assumptions

In loosely coupled interoperability, it is assumed that the role-based interorganizational workflow of the construction project conforms to the following characteristics.

(1) The intraorganizational workflows (the parts of workflows that are associated with other organizations) of participant organizations are processed iteratively in parallel.
(2) The correlation parameters of interorganizational workflows (matrix R) do not vary with time.
(3) The reprocessed workflow for each organization is a function of the workflow processing in the previous iteration stage.

In this study, the Design Iteration Model [31] was used as a reference for the analysis of the role-based interorganizational workflow of the construction project. In this model, the Work Transfer Matrix (WTM) is an extension to the DSM that contains additional numerical information. The WTM contains two separate sets of matrix data: an off-diagonal matrix of "Strength of Dependence Measures" whose elements indicate the amount of rework done during the iteration process and a diagonal matrix of "Task Times" that contains the time taken for each task.

In the analysis of role-based interorganizational workflows, the matrix R, which represents the correlations of interorganizational workflows, is similar to the off-diagonal matrix, which represents the

"Strength of Dependence Measures" in the WTM. The time matrix, W, (a diagonal matrix) denotes the time required for the intraorganizational workflow of each organization, i.e., the time that is normally required for each organization to complete its intraorganizational workflow under the condition of complete information. The time matrix is similar to the diagonal matrix of "Task Times" in the WTM.

### 3.2.2. Workflow Performance Indicators

In the iteration process for role-based interorganizational workflows, each iteration is characterized by a workflow vector $u_t$. This vector indicates how much of the intraorganizational workflow for each organization is processed in the *r*th iteration. The initial workflow vector $u_0$ is a vector of ls, which indicates that the intraorganizational workflows of all organizations need to be processed during the first iteration. During each iteration, the workflow of each organization is created for the next iteration, according to the linear rule.

In each iteration, the workflow reprocessing workload of each organization is determined using matrix R, as follows.

$$u_{t+1} = R{\cdot}u_t. \tag{2}$$

The workflow vector $u_t$ is expressed as:

$$u_t = R^t{\cdot}u_0. \tag{3}$$

The total amount of workflow processed during the iteration process is $U$, i.e., the sum of all the workflow vectors.

$$U = \sum_{t=0}^{\infty} u_t \tag{4}$$

For matrix R, there are linearly independent vectors.

$$R = S{\cdot}\Lambda{\cdot}S^{-1} \tag{5}$$

where $\Lambda$ is a diagonal matrix whose elements are eigenvalues of matrix R, and it is called an eigenstructure matrix. $S$ is a matrix composed of eigenvectors corresponding to eigenvalues of matrix R, and is called an eigenvector matrix.

$$R^t = S{\cdot}\Lambda^t{\cdot}S^{-1} \tag{6}$$

The total workflow vector U can, therefore, be expressed by the equation below.

$$U = S\left(\sum_{t=0}^{M} \Lambda^t\right)S^{-1}u_0 \tag{7}$$

If the magnitude of the maximum eigenvalue is less than one, then the interorganizational workflow process will converge (i.e., as *M* increases to infinity, the total workflow vector *U* remains bounded). An eigenvalue greater than one corresponds to a design process in which, if an organization performs one unit of workflow processing during an iteration stage, it will create more than one unit of workflow processing for that organization at a future stage. Such a system is unstable, and the vector *U* will not converge. Instead, it will grow without bounds as *M* increases. If we take the limit as *M* approaches infinity, we can use the formula below.

$$\lim_{M \to \infty} \sum_{t=0}^{M} \Lambda^t = (I - \Lambda)^{-1} \tag{8}$$

where I is a unit matrix, *U* could be written as the equation below.

$$U = S(I - \Lambda)^{-1}S^{-1}u_0 \tag{9}$$

The total workflow vector $U$ is multiplied by the time matrix $W$ in order to obtain the total amount of time spent on the workflow of each organization, where $W$ is a diagonal matrix. The vector A represents the amount of time spent on the workflow of each organization during the iteration process.

$$A = WU = W \sum_{t=0}^{\infty} u_t. \tag{10}$$

The vector $A$ can also be written as the equation below.

$$A = W \sum_{t=0}^{\infty} R^t \cdot u_0. \tag{11}$$

or if the maximum eigenvalue of $A$ is less than 1, Equation (11) can be simplified to the formula below.

$$A = W(I - R)^{-1} u_0. \tag{12}$$

The total time taken to complete the workflow processing of each organization ($T$) is the sum of the time taken for each iteration stage. The time spent on each iteration stage is the longest time taken for workflow processing of any organization in that stage.

$$T = \sum_{t=0}^{\infty} \max_i [W u_t]^{(i)} \tag{13}$$

where $[*]^{(i)}$ is the $i$th element of the vector.

Effort ($E$) is considered to be the total time spent on the interorganizational workflow process. It is the sum of all the time spent on the workflow processing of the individual organizations.

$$E = \sum_{i=1}^{n} \left[ w \sum_{t=0}^{\infty} R^t u_0 \right]^{(i)} = \sum_{i=1}^{n} A^{(i)} \tag{14}$$

The variables $T$ and $E$ are important quantities for managing role-based interorganizational workflow processes. In the design phase of a prefabricated building project, time is an important determining factor of the design process, while effort is an indicator of the design cost.

## 4. Analysis Example: A Role-Based Interorganizational Workflow for a Construction Project

Taking the detailed design stage of a prefabricated building project as an example, this paper regards the intraorganizational workflow for each organization as a "task." Using the Design Iteration Model as a reference, the role-based interorganizational workflow of a construction project was analyzed using loosely coupled interoperability.

*4.1. Determination of Correlation Matrix and Time Matrix*

4.1.1. Determination of the Correlation Matrix R

Fuzzy Analytic Hierarchy Process (FAHP) was applied to obtain the numerical Design Structure Matrices (NDSMs) of downstream sensitivity and the probability of change from the BDSM shown in Figure 3. The resulting NDSMs are shown in Figure 6a,b. The product of the elements of the downstream sensitivity and probability of change in NDSMs represents the correlation of the interorganizational workflow. Additionally, the matrix R was obtained, as shown in Figure 7.

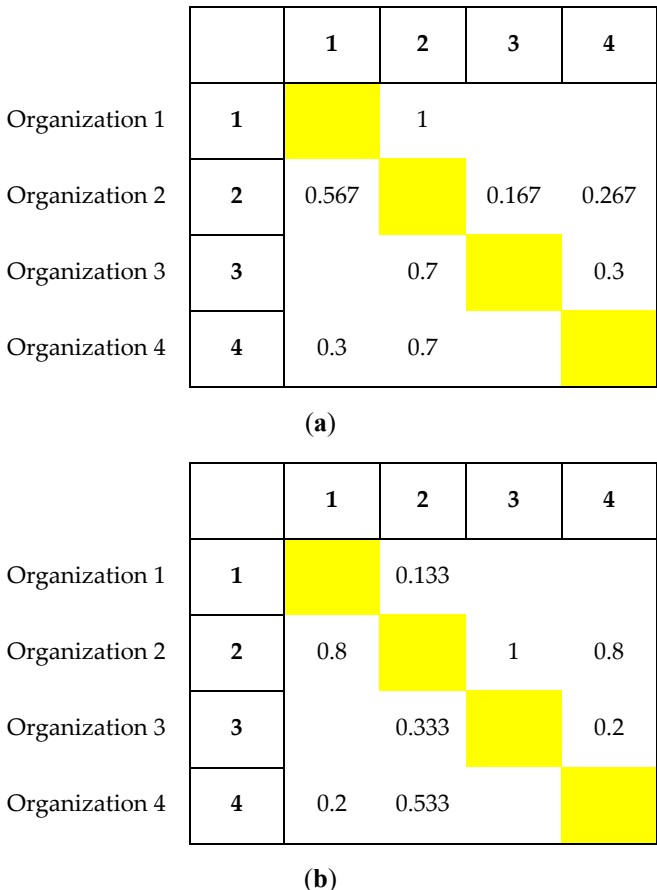

(**a**)

(**b**)

**Figure 6.** Numerical Design Structure Matrices (NDSMs) of downstream sensitivity (**a**) and probability of change (**b**).

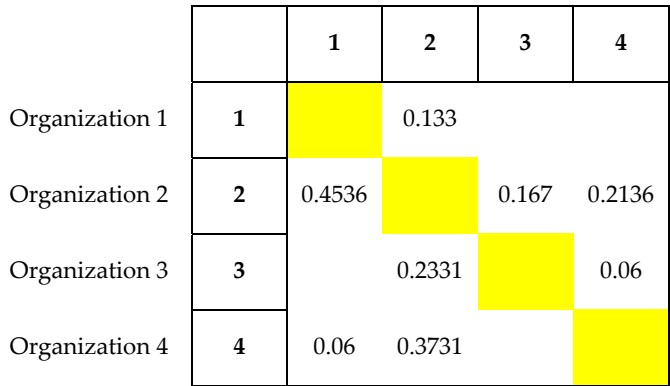

**Figure 7.** Matrix R showing the correlation between interorganizational workflows.

4.1.2. Determination of the Time Matrix

The time matrix, W, represents the time required for each organization to complete their part of the intraorganizational workflow under the condition of complete information. The diagonal elements of the time matrix are the times that are normally required for each organization to complete workflow

processing. As can be seen from the diagonal elements of the following time matrix W, the component design team of Organization 2 takes two units of time (i.e., weeks) to complete its workflow.

$$
W = \begin{bmatrix} 3 & & & \\ & 2 & & \\ & & 0.5 & \\ & & & 1.5 \end{bmatrix}
\tag{15}
$$

*4.2. Calculation of Workflow Performance Indicators*

For the example of a prefabricated building project, the role-based interorganizational workflow is analyzed, according to matrix *R* in Figure 7.

$$
R = \begin{bmatrix} 0 & 0.133 & 0 & 0 \\ 0.4536 & 0 & 0.167 & 0.2136 \\ 0 & 0.2331 & 0 & 0.06 \\ 0.06 & 0.3731 & 0 & 0 \end{bmatrix}
\tag{16}
$$

The matrix *R* is decomposed. The eigenvalue ($\Lambda$) and eigenvector (S) matrices are shown below.

$$
\Lambda = \begin{bmatrix} 0.4379 & & & \\ & -0.0152 + 0.0147i & & \\ & & -0.0152 - 0.0147i & \\ & & & -0.4075 \end{bmatrix}
\tag{17}
$$

$$
S = \begin{bmatrix} 0.1994 & -0.1643 - 0.1111i & -0.1643 + 0.1111i & 0.2276 \\ 0.6566 & 0.0311 - 0.0055i & 0.0311 + 0.0055i & -0.6973 \\ 0.4299 & 0.8853 + 0.0000i & 0.8853 - 0.0000i & 0.3098 \\ 0.5867 & -0.3452 + 0.2384i & -0.3452 - 0.2384i & 0.6049 \end{bmatrix}
\tag{18}
$$

Since the maximum eigenvalue ($\max(|a_{ij}|)$) < 1, the iteration of the role-based interorganizational workflow is convergent.

4.2.1. Calculation of the Total Time Spent on the Interorganizational Workflow Process (*E*)

The total amount of time spent on the interorganizational workflow for each organization is shown below.

$$
A = W(I - R)^{-1} u_0
$$

$$
= \begin{bmatrix} 3 & & & \\ & 2 & & \\ & & 0.5 & \\ & & & 1.5 \end{bmatrix} \begin{bmatrix} 1.0762 & 0.1631 & 0.0272 & 0.0365 \\ 0.5727 & 1.2262 & 0.2048 & 0.2742 \\ 0.1502 & 0.3139 & 1.0524 & 0.1302 \\ 0.2782 & 0.4673 & 0.0780 & 1.1045 \end{bmatrix} \begin{bmatrix} 1 \\ 1 \\ 1 \\ 1 \end{bmatrix}
\tag{19}
$$

$$
= (3.9090, \ 4.5558, \ 0.8234, \ 2.8920)^T
$$

The effort is the total amount of time spent on the interorganizational workflow process.

$$
E = \sum\nolimits_{i=1}^{n} A^{(i)} = 3.9090 + 4.5558 + 0.8234 + 2.8920 = 12.1802
\tag{20}
$$

From the above results, it can be seen that:

(1) The effort for the workflow processing workload (4.5558) and that for the workflow vector (2.2779) of Organization 2 (component design team) are the largest, mainly due to the fact that the workflow of this organization is closely correlated with those of other organizations, which requires repeated workflow iteration with other organizations;

(2) The effort for the workflow vector of Organization 4 (construction planning team) is the second largest (1.9280). The workflow of Organization 4 is moderately correlated with the workflows of other organizations.

4.2.2. Calculation of the Total Time Taken to Complete the Workflow Processing of Each Organization (*T*)

In the iterative processing of role-based interorganizational workflows, the time taken for each organization to process the initial workflow is shown below.

$$\mathrm{w}u_0 = \begin{bmatrix} 3 & & & \\ & 2 & & \\ & & 0.5 & \\ & & & 1.5 \end{bmatrix} \begin{bmatrix} 1 \\ 1 \\ 1 \\ 1 \end{bmatrix} = \begin{bmatrix} 3* \\ 2 \\ 0.5 \\ 1.5 \end{bmatrix} \tag{21}$$

In the first iteration, the workflow processing time of each organization is shown below.

$$\mathrm{w}u_1 = (0.3990,\ 1.6684*,\ 0.1466,\ 0.6496)^T \tag{22}$$

Similarly, the workflow processing time of each organization in the *r*th iteration can be calculated as:

$$\mathrm{w}u_3 = (0.0805,\ 0.3106*,\ 0.0332,\ 0.1229)^T \tag{23}$$

$$\mathrm{w}u_8 = (0.0022,\ 0.0032*,\ 0.0007,\ 0.0032)^T \tag{24}$$

$$\mathrm{w}u_{11} = (0.0002,\ 0.0005*,\ 0.0000,\ 0.0002)^T \tag{25}$$

$$\mathrm{w}u_{13} = (0.0000,\ 0.0000*,\ 0.0000,\ 0.0000)^T. \tag{26}$$

$$\mathrm{T} = \sum_{k=0}^{\infty} \max_i [Wu_k]^{(i)} = 3 + 1.6684 + 0.4788 + 0.3106 + 0.0893 + 0.0579$$
$$+ 0.0167 + 0.0109 + 0.0032 + 0.0021 + 0.0006 + 0.0005 + 0.0002 + 0.0000 = 5.6392. \tag{27}$$

## 5. Comparison with Interorganizational Workflow under Case Transfer Interoperability

### 5.1. Assumptions

In loosely coupled interoperability, the analysis of role-based interorganizational workflow assumes that the "work set" of each organization is iterated in parallel among organizations. Similarly, case transfer interoperability can also be used in role-based interorganizational workflows. In case of transfer interoperability, the interorganizational workflow for TMOs of a construction project is a series of connected workflows of the participant organization. We assume that this form of interorganizational workflow is equal to the sequential iteration of workflows among participant organizations, i.e., the sequential iteration of the "work set" of each organization.

The Sequential Iteration Model in engineering design assumes that all coupled tasks are executed sequentially and that the time taken for tasks and the redo probability of tasks are constant. Then, the Markov chain method is used to analyze and model the process of tasks iteration, and the execution time of tasks in a serial mode is calculated [32]. The Sequential Iteration Model can be used as a reference for interorganizational workflow in the case of transfer interoperability. That is, the "work set" of each organization (the part of the intraorganizational workflow which is associated with other organizations) can be used for modeling and analysis, instead of the "task."

### 5.2. Calculation of Workflow Performance Indicators in Case Transfer Interoperability

Taking the above prefabricated building project as an example, the case transfer interorganizational workflow was analyzed based on the Sequential Iteration Model. First, the matrix R for the

interorganizational workflow correlation shown in Figure 7 and the time matrix were merged into a DSM matrix, *M*, which is a Sequential Iteration Design Structure Matrix.

$$M = \begin{bmatrix} 3 & 0.133 & 0 & 0 \\ 0.4536 & 2 & 0.167 & 0.2136 \\ 0 & 0.2331 & 0.5 & 0.06 \\ 0.06 & 0.3731 & 0 & 1.5 \end{bmatrix} \tag{28}$$

Taking the ordering, "Organization 1, Organization 2, Organization 3, Organization 4", of workflow sequential iteration as an example, the Markov chain for processing role-based interorganizational workflows can be divided into the following four stages.

(1) Stage 1: intraorganizational workflow processing for Organization 1,
(2) Stage 2: transferring to the workflow processing of Organization 2 and performing iterative processing between Organization 2 and Organization 1,
(3) Stage 3: transferring to the workflow processing of Organization 3 and performing iterative processing between Organization 3 and Organization 1, and Organization 2,
(4) Stage 4: transferring to the workflow processing of Organization 4 and performing iterative processing between Organization 4 and Organization 1, and Organization 2, and Organization 3.

In the case of transfer interoperability, due to the sequential processing mode among organizations, only one participant organization separately processes the project workflow at any given time. Therefore, the total time taken to complete the workflow processing of each organization in the case of transfer interoperability ($T^*$) is the same as the total time spent on the interorganizational workflow process in the case of transfer interoperability ($E^*$).

$$T^* = E^* = T_1 + T_2 + T_3 + T_4 = 3 + 2.5529 + 0.9665 + 2.3798 = 8.8992 \tag{29}$$

In the Markov chain of the interorganizational workflow processing, there are $4 \times 3 \times 2 \times 1 = 24$ kinds of ordering due to the different iteration sequencing of workflows among organizations. Among the 24 kinds of ordering, there are two kinds of ordering for which $T^*$ and $E^*$ are the lowest—namely, the ordering "Organization 1, Organization 3, Organization 2, Organization 4" and "Organization 3, Organization 1, Organization 2, Organization 4."

$$T^*_{min} = E^*_{min} = 8.6727. \tag{30}$$

The values of $T^*$ and $E^*$ are the largest for the ordering "Organization 4, Organization 2, Organization 3, and Organization 1."

$$T^*_{max} = E^*_{max} = 10.1591. \tag{31}$$

Therefore, for the role-based interorganizational workflow, it is suggested to adopt the ordering "Organization 1, Organization 3, Organization 2, Organization 4" or "Organization 3, Organization 1, Organization 2, Organization 4" in case of transfer interoperability.

*5.3. Contrastive Analysis*

The value of T in loosely coupled interoperability is lower than $T^*_{min}$ in the case of transfer interoperability. The ratio of $T/T^*_{min}$ (in %) is:

$$T/T^*_{min} \times 100\% = 5.6392/8.6727 \times 100\% = 65.02\%. \tag{32}$$

While E in loosely coupled interoperability is larger than $E^*_{min}$ in the case of transfer interoperability. The ratio of $E/E^*_{min}$ (in %) is shown below.

$$E/E^*_{min} \times 100\% = 12.1802/8.6727 \times 100\% = 140.44\%. \tag{33}$$

Thus, loosely coupled interoperability significantly reduces the value of T and increases the value of E. Additionally, role-based interorganizational workflows for both case transfer and loosely coupled interoperability were analyzed and compared. The results are shown in Table 1.

**Table 1.** Contrastive analysis of role-based interorganizational workflows for two forms of interoperability: case transfer and loosely coupled.

| Interorganizational Workflow | Case Transfer | Loosely Coupled |
|---|---|---|
| Requirements for interorganizational information system | Low requirements | High requirements |
| Iteration method | Sequential iteration of inter-organization | Parallel iteration of inter-organization |
| Number of iterations | More | Less |
| Iteration range | From small range iteration among part organizations to whole range iteration among temporary multi-organizations (TMOs) | Whole range iteration among TMOs |
| Total time taken to complete the workflow processing of each organization (T) | Long | Short |
| Total time spent on the inter-organizational workflow process (E) | Small | Big |

## 6. Discussion

The analysis of role-based interorganizational workflows only considers the part of an intraorganizational workflow that is associated with other organizations. For TMOs of construction projects, this part of the workflow of each organization is regarded as a "task." Interorganizational workflows are treated with parallel iteration among different "tasks," i.e., the parallel iteration of "work sets" of each organization among TMOs. Combined with workflow interaction among organizations, a simplified and easy-to-understand model for the analysis of interorganizational workflows based on participant organization was developed using the Design Iteration Model as a reference. After several iterations, the performance indicators T and E for interorganizational workflows were obtained. This model can reflect the iteration of work processes among different organizations due to the workflow interaction between organizations.

### 6.1. Suggestions

Based on the results of this study, the following suggestions are made.

(1) A perfect BIM collaborative work platform (an interorganizational information system) should be established to achieve the integration of TMOs. Participant organizations should more effectively and frequently exchange information. A good interorganizational information exchange scheme would detect all kinds of work conflicts in a timely manner, and would, thus, reduce the iteration frequency and degree of the interorganizational workflow.

(2) A coordinator for interorganizational workflow management is proposed. Interorganizational interfaces management should be improved. Additionally, various barriers to cooperation between organizations should be reduced. The proper specification of interorganizational interfaces reduces the need for interactions between organizations.

(3) The values of T and E for interorganizational workflows for TMOs would be reduced by reducing the proportion of relevant interorganizational workflows and increasing the proportion of workflows that are independent of other organizations. For example, basic principles, rules, and concepts of relevant interorganizational work should be clarified before implementing interorganizational cooperation so as to facilitate the implementation of each organization's work. Furthermore, standardized design and modular design should be adapted in prefabricated building projects in order to reduce the correlation between interorganizational workflows.

(4) Participant organizations should cooperate with each other so that upstream organizations in the workflow can consider related problems of downstream organizations as soon as possible and, thus, give timely feedback. This would reduce the probability of change in downstream organizations caused by workflow processing in upstream organizations. Moreover, due to the delivery of timely feedback, the downstream sensitivity of the downstream organization to the workflow processing of the upstream organization would be reduced.

## 6.2. Research Significance and Implications

With the rapid development of the Internet and e-commerce, virtual enterprise and outsourcing have become popular ways for modern enterprises to cope with economic globalization. Interorganizational workflow technologies are widely used by business enterprises for process re-engineering, process management, and process automation. At present, most research on interorganizational workflow focuses on workflow modeling, i.e., transforming business processes into executable computer languages. The aim of such research is to improve the Workflow Management System (WfMS), and research findings are seldom directly used to analyze and guide the practice of interorganizational process management.

In this study, interorganizational workflows were analyzed and compared from an organizational perspective in order to investigate interorganizational rework and iteration phenomena. The study mainly relates to participant organizations of TMOs in a construction project. However, the analysis methods and findings can be extended to other fields and other types of interorganizational collaborative workflow, e.g., in manufacturing, e-commerce, and e-government fields. Moreover, they are not only applicable to TMOs in construction projects, but also to participants of New Product Development (NPD), supply chain management, virtual enterprises, and more. The analysis methods and findings used in this study can also be used to analyze and guide the practice of interorganizational collaborative workflows and improve the performance of interorganizational process management. These analysis methods and findings can help to:

(1) Determine the appropriate interoperability form and implementation mode of participant-based interorganizational workflows by optimization and comparison.

(2) Analyze and calculate the time spent and workload for each participant in an interorganizational workflow process.

(3) Define measures and methods for improving interorganizational workflow management.

Additionally, the analysis methods and findings of this study can also help to develop a participant-oriented WfMS. This could be used to: develop tools for calculating the time spent and workload of participants in interorganizational workflows, develop a system with an optimization function for the implementation mode of interorganizational workflows, and produce a workflow implementation platform and management system for participants in an optimized implementation mode. The development of a participant-oriented WfMS is one of our research interests.

## 6.3. Future Research and Practice

Besides research studies related to WfMS whose aim is to process automation, future areas of study could also include related operations and practices of participant-based interorganizational workflows, with the aim of process re-engineering and process management.

In future research and practice, we plan to study the following:

(1)　A new mode that combines case transfer interoperability and loosely coupled interoperability, i.e., a mixed model of sequential and parallel iterations for interorganizational workflows with optimized ordering of workflow processing among organizations,

(2)　How to achieve centralized interorganizational workflows in capacity-sharing interoperability. Related research studies involve process design, implementation plans, support environments for interorganizational workflows, etc.,

(3)　Other related research, such as that related to overcoming barriers for interorganizational workflow, interaction mechanisms of interorganizational information flow, and the analysis of mechanisms of interorganizational co-evolution.

## 7. Conclusions

Construction projects are carried out by TMOs. With a mutual restriction and the interaction of many factors, complex workflow relationships are formed among TMOs. The implementation processes of construction projects are iterative processes involving continuous modification and improvement among participating organizations. Traditional workflow analysis methods for a single organization are not suitable for the study of the implementation processes of construction projects. Rather, this requires an interorganizational workflow analysis method, which is based on organizational roles, and is associated with the collaborative relationships between organizations. Current research on interorganizational workflows fails to take an organizational perspective, and ignores collaborative relationships between participants, such as rework and iteration performed by reciprocal workflows or interdependence and coupling among organizations.

The modeling of interorganizational workflows emphasizes links between the workflows of different organizations and the coordination of the interactions between organizations. In the present study, interorganizational workflows are analyzed based on an organization level, by taking participant organization as the role. In a loosely coupled form, the analysis of role-based interorganizational workflows assumes that the "work set" of each organization is iterated in parallel among organizations. The Design Iteration Model was used as a reference for the analysis. The performance indicators T and E were obtained for the interorganizational workflows. Furthermore, interorganizational workflows of the case transfer form are assumed to perform sequential iteration of workflows among participant organizations.

The contributions of this paper are as follows.

(1)　From an organizational perspective, a role-based interorganizational workflow analysis model is proposed with the purpose of investigating interorganizational rework and iteration phenomena for TMOs in a construction project. Additionally, performance indicators of the interorganizational workflow are obtained from the modeling.

(2)　Based on a contrastive analysis of role-based interorganizational workflows in two forms of interoperability, loosely coupled interoperability was found to significantly reduce the value of T of the interorganizational workflow, but increase its value of E, compared with case transfer interoperability.

(3)　Some suggestions are put forward to improve interorganizational workflow management and reduce the values of T and E of inter-organizational workflows for TMOs in construction projects.

The findings of this paper are as follows.

(1)　Due to interorganizational rework and iteration phenomena, the workflow processing time and workload of participant organizations will increase. The extent of the increase is determined by the correlation between the workflows of different organizations.

(2) The more organizations that participate in an interactive workflow, the more complex repetition and iteration of work processes there are and the higher the corresponding time (or workload) indicator of the interorganizational workflow.

(3) The total time and workload of the interorganizational workflow can be reduced by reducing the correlation parameters of the workflow, such as downstream sensitivity ($s_{ij}$) and the probability of change ($c_{ij}$), or by reducing the proportion of relevant interorganizational workflow (collaborative workflow).

(4) The total time of the interorganizational workflow in loosely coupled form is less than that in the case of transfer form. However, the total workload of the workflow in loosely coupled form is more than that in the case transfer form.

This research is significant for enhancing the function of interorganizational workflow management, enriching workflow modeling theory, and promoting interorganizational workflow management.

**Author Contributions:** Conceptualization, methodology, and writing—original draft preparation: J.J. Validation and writing—review and editing: Y.Z. Supervision, J.C. Project administration and funding acquisition: R.Z.

**Funding:** The National Key R&D Program of China, grant 2016YFC0702107, supported this research.

**Conflicts of Interest:** The authors declare no conflict of interest.

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
