# Peer review of "Analysis and Comparison of Role-Based Interorganizational Workflows for a Construction Project"

_applsci, doi:10.3390/app9183667_

Round 1

Reviewer 1 Report

Improve the exposición of flows manager. You should go deeper into the possibility of establishing tools in new products. 

Many references are made to the system of prefabricated construction, but it omits the behavior with traditional construction systems. 

You have to look for more relationships with the BIM corporate work platform.

The development of research lines can be improved because we consider that there are few references to future lines of research.

The layout of the article can be improved, in order to speed up its reading.

Author Response

Response to Reviewer 1 Comments

Point 1: Improve the exposición of flows manager. You should go deeper into the possibility of establishing tools in new products. 

Response 1: The discussion section added: “6.2. Research significance and implications. The possibility of establishing tools in new products were mentioned, as follows:

6.2. Research significance and implications

With the rapid development of the Internet and e-commerce, virtual enterprise and outsourcing have become popular ways for modern enterprises to cope with economic globalization. Interorganizational workflow technologies are widely used by business enterprises for process re-engineering, process management, and process automation. At present, most research on interorganizational workflow focuses on workflow modeling, i.e., transforming business processes into executable computer languages. The aim of such research is to improve the Workflow Management System (WfMS), and research findings are seldom directly used to analyze and guide the practice of interorganizational process management.

In this study, interorganizational workflows were analyzed and compared from an organizational perspective in order to investigate interorganizational rework and iteration phenomena. The study mainly relates to participant organizations of TMOs in a construction project. However, the analysis methods and findings can be extended to other fields and other types of interorganizational collaborative workflow, e.g., in manufacturing, e-commerce, and e-government fields; moreover, they are not only applicable to TMOs in construction projects, but also to participants of New Product Development (NPD), supply chain management, virtual enterprises, etc. The analysis methods and findings used in this study can also be used to analyze and guide the practice of interorganizational collaborative workflows and improve the performance of interorganizational process management. These analysis methods and findings can help to:

(1) Determine the appropriate interoperability form and implementation mode of participant-based interorganizational workflows by optimization and comparison;

(2) Analyze and calculate the time spent and workload for each participant in an interorganizational workflow process;

(3) Define measures and methods for improving interorganizational workflow management.

Additionally, the analysis methods and findings of this study can also help to develop a participant-oriented WfMS. This could be used to: develop tools for calculating the time spent and workload of participants in interorganizational workflows; develop a system with an optimization function for the implementation mode of interorganizational workflows; and produce a workflow implementation platform and management system for participants in optimized implementation mode. The development of a participant-oriented WfMS is one of our research interests.

Point 2: Many references are made to the system of prefabricated construction, but it omits the behavior with traditional construction systems. 

Response 2: It is my carelessness. Sorry. An example of traditional construction replaced the example of prefabricated building project in “1. Introduction”.  It added: “For example, such projects involve many participant organizations, such as the owner, designer, constructor, supplier, general contractor, subcontractor, operator, etc.”

In order to better reflect correlation between the interorganizational workflows for participant organizations, the paper has taken prefabricated building project as examples. But the analysis methods and findings are applicable for TMOs of traditional construction systems.

Point 3: You have to look for more relationships with the BIM corporate work platform.

Response 3: Sorry for a mistake in writing. “BIM Cooperative Work Platform” has been replaced with “BIM collaborative work platform”. There are some introductions for “BIM collaborative work platform” in “2. Modeling of role-based interorganizational workflows for construction projects”, as follows:

Each participant organization interoperates in the interorganizational workflow using a Building Information Modeling (BIM) collaborative work platform (an interorganizational information system) to integrate TMOs. … Multiple participant organizations coordinate with the interorganizational workflow based on the BIM collaborative work platform. Through multi-organizational cooperation and interorganizational workflow interoperation, component problems can be modified iteratively before manufacture or installation using a BIM integrated model so as to achieve efficient coordination among component design, factory manufacturing, and site installation. Additionally, negotiation and coordination with cooperative work can reduce the time and effort involved in communication among participant organizations.

Additionally, a description was added:

BIM collaborative work platforms—for example, the “BIM Collaborative Platform of Project Management” developed by Glodon Company, Ltd., China—can achieve the integration and cooperation of participant organizations such as the owner, designer, fabricator, and erector.

Point 4: The development of research lines can be improved because we consider that there are few references to future lines of research.

Response 4: The discussion section added: 6.3. Future research and practice.

6.3. Future research and practice

Besides researches related to WfMS whose aim is to process automation, future areas of study could also include related operations and practices of participant-based interorganizational workflows, with the aim of process re-engineering and process management.

In future research and practice, we plan to study the following:

(1) A new mode which combines case transfer interoperability and loosely coupled interoperability, i.e., a mixed model of sequential and parallel iterations for interorganizational workflows with optimized ordering of workflow processing among organizations;

(2) How to achieve centralized interorganizational workflows in capacity-sharing interoperability. Related researches involve process design, implementation plans, support environments for interorganizational workflows, etc.;

(3) Other related research, such as that related to overcoming barriers to interorganizational workflow, interaction mechanisms of interorganizational information flow, and the analysis of mechanisms of interorganizational co-evolution.

Point 5: The layout of the article can be improved, in order to speed up its reading.

Response 5: The layout of the article improved.

Thank you for really valuable comments and suggestions for authors.

Best regards.

Reviewer 2 Report

- “A method between macro and micro for participant organizations is proposed…”: This sentence in the abstract seems incomplete and does not make sense. Macro and Micro of what?

- “model is also made”… inappropriate choice of words for academic writing. “Made” should be replaced with “developed”. The authors are advised to get their manuscript proofread and edited in English to the a publishable standard. 

- The research/knowledge gap is not stated in the abstract. 

- The research aim is not clear in the introduction. 

- The reviewer expects to see a section named Research Methodology, where the methods adopted for the study are described and justified. The lack of such section is a major flaw. 

- The discussion section is week and must be improved and elaborated by bringing out the significance of the study results. The implications for practice must also be explained. 

- In the conclusions, what are the major findings of this study? 

Author Response

Response to Reviewer 2 Comments

Point 1: “A method between macro and micro for participant organizations is proposed…”: This sentence in the abstract seems incomplete and does not make sense. Macro and Micro of what?

Response 1: “A method between macro and micro for participant organizations is proposed…”:

This sentence in the abstract has been deleted。

Point 2: “model is also made”… inappropriate choice of words for academic writing. “Made” should be replaced with “developed”. The authors are advised to get their manuscript proofread and edited in English to the a publishable standard. 

Response 2: “Made” has been replaced with “developed”. The paper has undergone English language editing by MDPI.

Point 3: The research/knowledge gap is not stated in the abstract.

Response 3: The research/knowledge gap has been stated in the abstract, as follows:

Therefore, an interorganizational workflow analysis method based on organizational roles and associated with their collaborative relationships is required. 

Point 4: The research aim is not clear in the introduction. 

Response 4: The research aim has been rewritten in the introduction, as follows:

This paper studies the interorganizational workflow of TMOs from the perspective of organizational roles and their interaction. The research aims include the following:

the development of an analysis model of role-based interorganizational workflow; the calculation of time or workload performance indicators via modeling considering interorganizational reworking and iteration phenomena;

(3) the analysis and contrasting of interorganizational workflows with different forms of interoperability, so as to determine an appropriate form of interoperability for interorganizational workflows;

(4) the discussion of measures to improve interorganizational workflow management.

This study conducts a modeling-based analysis of interorganizational workflows of TMOs in construction projects from a role-based perspective and calculates performance indicators such as time or workload by modeling. Interorganizational workflows in a loosely coupled form are compared with interorganizational workflows in a case transfer form. Moreover, some suggestions are put forward to improve interorganizational workflow management.

Point 5: The reviewer expects to see a section named Research Methodology, where the methods adopted for the study are described and justified. The lack of such section is a major flaw. 

Response 5: A section named “3. Research methodology” was added by modifying previous paper. It includes: 3.1. Correlation parameters for interorganizational workflow; 3.2. Analysis of role-based interorganizational workflow.

Some content of  “3. Research methodology”  are as follows:

In this study, a loosely coupled interorganizational workflow iteration model was developed referring to the Design Iteration Model (an engineering design model). This was achieved by taking organization as a role and aiming at interorganizational rework and iteration phenomena caused by interactive workflows together with interdependence and coupling relationships among participant organizations. The analysis was based on each participant organization.

The Design Iteration Model was developed using the Design Structure Matrix (DSM) [31]. This model assumes that all coupled tasks are executed simultaneously in parallel and that iteration workload is constant. Iteration-driven tasks in the design process are identified by establishing a Work Transfer Matrix (WTM) model. The Design Iteration Model was used for reference in the analysis of a role-based interorganizational workflow of a construction project.

Point 6: The discussion section is weak and must be improved and elaborated by bringing out the significance of the study results. The implications for practice must also be explained.  

Response 6: The discussion section added:

6.2. Research significance and implications;

6.3. Future research and practice.

The paper brought out the significance of the study results, and explained the implications for practice.  

6.2. Research significance and implications

With the rapid development of the Internet and e-commerce, virtual enterprise and outsourcing have become popular ways for modern enterprises to cope with economic globalization. Interorganizational workflow technologies are widely used by business enterprises for process re-engineering, process management, and process automation. At present, most research on interorganizational workflow focuses on workflow modeling, i.e., transforming business processes into executable computer languages. The aim of such research is to improve the Workflow Management System (WfMS), and research findings are seldom directly used to analyze and guide the practice of interorganizational process management.

In this study, interorganizational workflows were analyzed and compared from an organizational perspective in order to investigate interorganizational rework and iteration phenomena. The study mainly relates to participant organizations of TMOs in a construction project. However, the analysis methods and findings can be extended to other fields and other types of interorganizational collaborative workflow, e.g., in manufacturing, e-commerce, and e-government fields; moreover, they are not only applicable to TMOs in construction projects, but also to participants of New Product Development (NPD), supply chain management, virtual enterprises, etc. The analysis methods and findings used in this study can also be used to analyze and guide the practice of interorganizational collaborative workflows and improve the performance of interorganizational process management. These analysis methods and findings can help to:

(1) Determine the appropriate interoperability form and implementation mode of participant-based interorganizational workflows by optimization and comparison;

(2) Analyze and calculate the time spent and workload for each participant in an interorganizational workflow process;

(3) Define measures and methods for improving interorganizational workflow management.

Additionally, the analysis methods and findings of this study can also help to develop a participant-oriented WfMS. This could be used to: develop tools for calculating the time spent and workload of participants in interorganizational workflows; develop a system with an optimization function for the implementation mode of interorganizational workflows; and produce a workflow implementation platform and management system for participants in optimized implementation mode. The development of a participant-oriented WfMS is one of our research interests.

6.3. Future research and practice

Besides researches related to WfMS whose aim is to process automation, future areas of study could also include related operations and practices of participant-based interorganizational workflows, with the aim of process re-engineering and process management.

In future research and practice, we plan to study the following:

(1) A new mode which combines case transfer interoperability and loosely coupled interoperability, i.e., a mixed model of sequential and parallel iterations for interorganizational workflows with optimized ordering of workflow processing among organizations;

(2) How to achieve centralized interorganizational workflows in capacity-sharing interoperability. Related researches involve process design, implementation plans, support environments for interorganizational workflows, etc.;

(3) Other related research, such as that related to overcoming barriers to interorganizational workflow, interaction mechanisms of interorganizational information flow, and the analysis of mechanisms of interorganizational co-evolution.

Point 7: In the conclusions, what are the major findings of this study? 

Response 7: In the conclusions, the major findings of this study were discussed.

The findings of this paper are as follows:

(1) Due to interorganizational rework and iteration phenomena, the workflow processing time and workload of participant organizations will increase. The extent of the increase is determined by the correlation between the workflows of different organizations;

(2) The more organizations that participate in an interactive workflow, the more complex repetition and iteration of work processes there are and the higher the corresponding time (or workload) indicator of the interorganizational workflow;

(3) The total time and workload of the interorganizational workflow can be reduced by reducing the correlation parameters of the workflow, namely, downstream sensitivity ( ) and the probability of change ( ), or by reducing the proportion of relevant interorganizational workflow (collaborative workflow);

(4) The total time of the interorganizational workflow in loosely coupled form is less than that in case transfer form, however the total workload of the workflow in loosely coupled form is more than that in case transfer form.

Thank you for really valuable comments and suggestions for authors.

Best regards.

Round 2

Reviewer 1 Report

Now the work is correct. Congratulations on your corrections.

Reviewer 2 Report

Thanks to the authors for addressing the review comments.